# Emotional Exhaustion, a Proxy for Burnout, Is Associated with Sleep Health in French Healthcare Workers without Anxiety or Depressive Symptoms: A Cross-Sectional Study

**DOI:** 10.3390/jcm12051895

**Published:** 2023-02-27

**Authors:** Julien Coelho, Jacques Taillard, Adèle Bernard, Régis Lopez, Guillaume Fond, Laurent Boyer, Guillaume Lucas, François Alla, Daniel J. Buysse, Meredith L. Wallace, Catherine Verdun-Esquer, Pierre-Alexis Geoffroy, Emmanuel d’Incau, Pierre Philip, Jean-Arthur Micoulaud-Franchi

**Affiliations:** 1SANPSY, UMR 6033, University of Bordeaux, F-33000 Bordeaux, France; 2SANPSY, UMR 6033, CNRS, F-33000 Bordeaux, France; 3Service Universitaire de Médecine du Sommeil, CHU Bordeaux, F-33000 Bordeaux, France; 4Department of Neurology, Sleep Disorders Center, Gui-de-Chauliac Hospital, CHU Montpellier, F-34000 Montpellier, France; 5Inserm, U1061, Université Montpellier 1, F-34000 Montpellier, France; 6CEReSS-Health Service Research and Quality of Life Center, Assistance Publique des Hôpitaux de Marseille, Aix-Marseille University, 27, Boulevard Jean -Moulin, F-13000 Marseille, France; 7Fondation FondaMental, F-94000 Créteil, France; 8Pôle de Santé Publique, CHU Bordeaux, F-33000 Bordeaux, France; 9Department of Psychiatry, University of Pittsburgh, Pittsburgh, PA 15260, USA; 10Département de Psychiatrie et D’addictologie, AP-HP, GHU Paris Nord, DMU Neurosciences, Hopital Bichat—Claude Bernard, F-75018 Paris, France; 11GHU Paris—Psychiatry & Neurosciences, 1 Rue Cabanis, F-75014 Paris, France; 12Inserm, FHU I2-D2, Université de Paris, NeuroDiderot, F-75019 Paris, France

**Keywords:** sleep health, burnout, healthcare workers, cross-sectional

## Abstract

Burnout is frequent among healthcare workers, and sleep problems are suspected risk factors. The sleep health framework provides a new approach to the promotion of sleep as a health benefit. The aim of this study was to assess good sleep health in a large sample of healthcare workers and to investigate its relationship with the absence of burnout among healthcare workers while considering anxiety and depressive symptoms. A cross-sectional Internet-based survey of French healthcare workers was conducted in summer 2020, at the end of the first COVID-19 lockdown in France (March to May 2020). Sleep health was assessed using the RU-SATED v2.0 scale (RegUlarity, Satisfaction, Alertness, Timing, Efficiency, Duration). Emotional exhaustion was used as a proxy for overall burnout. Of 1069 participating French healthcare workers, 474 (44.3%) reported good sleep health (RU-SATED > 8) and 143 (13.4%) reported emotional exhaustion. Males and nurses had a lower likelihood of emotional exhaustion than females and physicians, respectively. Good sleep health was associated with a 2.5-fold lower likelihood of emotional exhaustion and associations persisted among healthcare workers without significant anxiety and depressive symptoms. Longitudinal studies are needed to explore the preventive role of sleep health promotion in terms of the reduction in burnout risk.

## 1. Introduction

Burnout, first described by Maslach et al. [1], is a state of psychological, emotional, and physical stress in response to prolonged exposure to an occupational trigger. It encompasses feelings of emotional exhaustion (EE, depletion of emotional resources), depersonalization (developing cynical attitudes about others, such as patients), and reduced professional accomplishment (negative evaluation of oneself). Burnout is related to adverse health outcomes, including mortality, cardiometabolic morbidity, depression and suicide, cognitive impairment, and reduced work performance [2]. The frequency of burnout among healthcare workers (HCWs) is high [3]. Recent meta-analyses yielded 6.0% and 11.2% burnout prevalence rates among physicians and nurses, respectively, worldwide [4,5]. Indeed, HCWs are consistently subjected to emotionally draining stressors in the provision of complex patient care and treatment [6], including heavy workloads, time pressure, and conflicts of value associated with low rewards, contradictory demands, and the lack of resources [7]. For instance, during the period of work overload due to the 2019 coronavirus (COVID-19) pandemic, HCWs reported increased exhaustion, anxiety, and depressive symptoms, as well as insomnia [8]. Sleep alterations due to the disturbance of sleep–wake rhythms by stress and work conditions (i.e., atypical work schedule with frequent night shifts and on-call duties) are also implicated in the development of burnout in HCWs [9,10]. 

The COVID-19 health crisis aggravated HCWs’ critical situation by massively disrupting their sleep [11] and mental health [8,12], worsening their work conditions, and increasing the prevalence of burnout [13]. Beyond facilitating the study of sleep alterations and their causes and consequences, the sleep health framework provides a new approach to the promotion of sleep as a positive dimension for the benefit of health [14]. It constitutes a privileged dimension for interventions that attempt to improve the sleep and workplace well-being of HCWs. A short, practical self-reported scale (RU-SATED) was recently developed and validated in a variety of languages [15,16,17,18,19], and it is a reliable and valid tool that has shown promise for the rapid evaluation of sleep health. It is a six-item self-reported questionnaire exploring the six sleep health dimensions (“Regularity”, “Satisfaction”, “Alertness”, “Timing”, “Efficiency”, and “Duration”). Good sleep health has been associated with several positive health outcomes, such as good cardiometabolic markers [20], a low risk of motor vehicle accidents [18], good physical [21] and mental health [22], and good self-perceived health status [23]. To date, good sleep health and its relationship with the absence of burnout have not been studied among HCWs. Burnout [24], anxiety, and depressive symptoms are closely associated with sleep [25]; therefore, they deserve particular attention in studies of the relationship between sleep and burnout. A study of 6307 firefighters revealed no interaction between insomnia and self-reported diagnoses of anxiety and depression in relation to EE [26], whereas a study conducted with 84 adults revealed an interaction between sleep quality and mental health in association with heart-rate variability, an objective indicator of chronic stress [27]. 

Thus, we used the RU-SATED questionnaire to (1) assess sleep health in a large population of HCWs (physicians and nurses) in a French public hospital at the end of the first COVID-19 lockdown (17 March–11 May 2020) and (2) investigate the relationship between the RU-SATED score and burnout. We hypothesized that good sleep health would be associated with a lesser likelihood of burnout. We also determined the consistency of the results among the six sleep health dimensions and evaluated the roles of anxiety and depressive symptoms in any such association.

## 2. Materials and Methods

This observational cross-sectional study was conducted from June to October 2020. The 8211 HCWs in Bordeaux University Hospital were asked to complete an Internet-based questionnaire. They were informed of the research objective (i.e., to evaluate sleep health). The STROBE (Strengthening the Reporting of Observational Studies in Epidemiology) statement was used to report this study [28].

The following sociodemographic data were collected: age, sex (male, female), job category (physician, nurse), and perceived socioeconomic status (comfortable, adequate, difficult).

The following data on work conditions, related to sleep–wake rhythms, were collected: work schedule (fixed or shift), night work (defined as >3 h work between 9 p.m. and 6 a.m. two times per week or more) [29], and telecommuting (yes, no). The study was conducted during the COVID-19 pandemic; therefore, we collected data concerning COVID-19 exposure (work in a COVID-19 unit; yes, no). 

Self-perceived health was assessed using a single item from the Minimum European Health Module [30]: “How is your health in general?” rated on a 4-point Likert scale ranging from 0 (“very good”) to 3 (“very bad”). We considered that participants who responded “bad” or “very bad” had poor self-perceived health. 

Anxiety and depressive symptoms were measured using the Patient Health Questionnaire-4, a reliable and valid four-item tool to which responses regarding the previous 2 weeks are provided on a 3-point Likert scale [31]. Responses to the first two items (feeling nervous, anxious, or on edge and not being able to stop or control worrying) are summed to obtain the anxiety score and those to the last two items (little interest or pleasure in doing things and feeling down, depressed, or hopeless) are summed to obtain the depressive symptom score. Cronbach’s alpha values for the anxiety and depressive symptom subscales were 0.82 and 0.81, respectively. Scores ≥ 3 were considered to indicate significant anxiety or depressive symptoms.

Sleep health was measured using the French version of the RU-SATED v2.0 scale [18], a self-administered questionnaire evaluating the six dimensions of sleep health: sleep “Regularity” (bedtimes and wake times occurring within a 1-h period across days), sleep “Satisfaction” (subjective assessment of “good” or “poor” sleep), “Alertness” during waking hours (awake all day without dozing), appropriate “Timing” (asleep or trying to sleep between 2:00 and 4:00 a.m.), “Efficiency” (<30 min wake time during the night), and adequate “Duration” (6–8 h/day). Items are rated on a 3-point Likert frequency scale (0, “rarely/never”; 1, “sometimes”; 2, “usually/always”). The total score, obtained by summing the item scores, ranges from 0 to 12; higher scores indicate better sleep health. In the absence of a validated cut-off, the total score was dichotomized based on the median, with scores > 8 indicating good sleep health, as in previous studies [23,32].

Emotional exhaustion, measured by the single-item version of the Maslach Burnout Inventory, was used as a proxy for overall burnout [33]. It consists of a single statement (“I feel burned out from my work”) rated on a 7-point Likert-scale ranging from “never” to “daily.” This item correlates strongly with the global score for EE (r ≥ 0.76), has good external validity in association with workplace well-being outcomes (suicidality, serious thoughts of dropping out, perceived major medical error) [34], and has greater predictive power for burnout than does depersonalization [35]. Responses of “once a week” or more frequently were considered to indicate the presence of EE as suggested in the original validation study [33]. Conversely, responses of “a few times per month” or less frequently were considered to indicate the absence of EE.

Descriptive statistics (frequencies for categorical variables, means, and standard deviations for continuous variables) were calculated. Univariate associations with EE were assessed using the chi-squared test for categorical variables and Student’s *t*-test for continuous variable. Multivariable logistic regression analysis was performed to estimate adjusted odds ratios (ORs) with 95% confidence intervals (CIs) for the association between good sleep health (the main explanatory variable) and the absence of EE (the dependent variable). Odds ratios were displayed as >1 for ease of interpretation. We selected confounding factors according to a review of the literature for factors influencing both sleep and burnout (age, sex, job category, work schedule, night work, telecommuting, work in a COVID-19 unit, and self-perceived health) [13,36,37,38]. The perceived socioeconomic status was also included, as it was associated with sleep health and EE at the 0.20 threshold in univariate analyses. The linearity hypothesis for quantitative variables was verified using fractional polynomials [39], and analysis of the residuals was conducted to confirm the suitability of the model. To explore the dose–response relationship, a sensitivity analysis was performed with sleep health represented as a continuous variable in a linear regression model. Covariate interactions with sleep health were examined, and no result was significant at the 0.20 threshold [40]. Sleep health interacted significantly with anxiety (*p* = 0.152) and depressive symptoms (*p* = 0.037) in relation to EE. Thus, stratified analyses were conducted to explore whether the relationship between good sleep health and the absence of EE was persistent and similar in HCWs according to the presence or absence of anxiety and depressive symptoms.

To determine the consistency of associations among the six sleep health dimensions, we performed a secondary analysis of the association between each dimension and the absence of EE. For all tests, the accepted significance level was 5%. The data analyses were conducted using R v. 4.1.2.

## 3. Results

### 3.1. Characteristics of the Study Population

The sample comprised 1069 HCWs, or 13.0% of the 8211 HCWs at Bordeaux University Hospital. The mean age was 39.2 ± 11.3 years, 900 (84.2%) participants were female, and 866 (81.0%) participants were nurses. A total of 514 (48.1%) HCWs had shift schedules and 199 (18.6%) were engaged in night work. Anxiety and depressive symptoms were present in 384 (35.9%) and 167 (15.6%) HCWs, respectively. Further descriptive data are presented in Table 1.

The mean RU-SATED score was 7.9 ± 2.4, with 474 (44.3%) HCWs reporting good sleep health. Good sleep health was more common among physicians, those with comfortable socioeconomic status, and HCWs on fixed schedules. Conversely, poor sleep health was more common among night workers and those who worked in a COVID-19 unit and/or had poor self-perceived health, anxiety, and/or depressive symptoms.

A total of 143 (13.4%) HCWs reported EE. This condition was more common among females, those in difficult socioeconomic status, and those with poor self-perceived health, anxiety, and/or depressive symptoms.

### 3.2. Association between EE and Sleep Health

HCWs reporting good sleep health had a two-fold lower likelihood of EE (*n* = 38 [8.0%]) than those reporting poor sleep health (*n* = 105 [17.6%]; OR = 2.46 [1.66–3.64], *p* < 0.001). In the analysis adjusted for age, sex, job category, perceived socioeconomic status, work schedule, night work, telecommuting, work in a COVID-19 unit, and self-perceived health, good sleep health remained associated with a lesser likelihood of EE (OR = 2.50 [1.61–3.88], *p* < 0.001). Males, nurses, and those with a comfortable socioeconomic status had lesser likelihoods of EE than females, physicians, and those with difficult socioeconomic status, respectively. Shift work and night work were not associated with EE in the multivariate models (Figure 1). Sensitivity analyses showed similar results using sleep health and EE as continuous variables (ß = −0.12 [−0.16; −0.08], *p* < 0.001; Appendix A).

### 3.3. Association between EE and Sleep Health According to ANXIETY and Depressive Symptoms

In adjusted analyses, EE was less common in HCWs with good sleep health among those without anxiety (*n* = 685; OR = 2.66 [1.20–5.89], *p* = 0.014) and without depressive symptoms (*n* = 902; OR = 2.56 [1.47–4.46], *p* < 0.001). Among HCWs with anxiety (*n* = 384) and those with depressive symptoms (*n* = 167), associations were not significant, and estimates were closer to 1 (OR = 1.52 [0.83–2.77], *p* = 0.166 and OR = 1.24 [0.50–3.12], *p* = 0.641, respectively; Figure 2). Sensitivity analyses showed similar results when including anxiety and depressive symptoms and their interaction term as covariates in the multivariate model (OR = 2.35 [1.10–5.04], *p* = 0.028; Appendix A). 

### 3.4. Associations between EE and Sleep Health Dimensions

In adjusted analyses, HCWs reporting that they were sometimes and usually/always satisfied with their sleep had two- and five-fold lesser likelihoods of EE, respectively, than those reporting that they were rarely/never satisfied. Similarly, HCWs reporting that they were usually/always alert had a two-fold lesser likelihood of EE than those reporting that they were rarely/never alert. The other sleep health dimensions were not associated with EE (Figure 3).

## 4. Discussion

### 4.1. Key Results

Good sleep health was associated with a two-fold lesser likelihood of EE, a proxy for burnout, during the COVID-19 health crisis in a sample of 1069 French HCWs. Males, nurses, and HCWs in comfortable socioeconomic status had lower likelihood of EE. None of these covariates explained the association between sleep health and EE in an adjusted analysis; neither did age, work schedule, night work, telecommuting, work in a COVID-19 unit, or self-perceived health. The sleep health dimensions of “Satisfaction” and “Alertness” were associated with low likelihoods of EE, as indicated by the at least five-fold lesser likelihood of EE among HCWs reporting that they were usually/always satisfied with their sleep than among those reporting that they were rarely/never satisfied. These associations were also significant among HCWs without anxiety or depressive symptoms. The overlap between burnout and depression [41], associated with decreased power in subgroup analyses, might explain the lack of a significant association among HCWs with anxiety or depressive symptoms.

### 4.2. Interpretation

Our findings are consistent with previous findings on relationships between sleep alterations and burnout [9,10]. The confirmation of this relationship using the RU-SATED scale was important from an interventional perspective. In the sleep health framework, sleep is a positive dimension including measurable sleep quality and modifiable sleep behaviors [14]. Thus, improvement of the sleep health of individuals, independently of the presence or absence of sleep disorders, may benefit their workplace well-being [14]. Work is an important contributor to the societal and social constraints that limit sleep opportunities for individuals [42,43]; thus, employers and HCWs should be aware of this framework and use it to prevent burnout. Nevertheless, further studies should consider other work characteristics, such as those measured by the Job Content Questionnaire (i.e., psychological demands, emotional demands, job control, social support) [44], that are important contributors to burnout [45]. Taking these numerous and highly correlated factors into account will likely require structural equation modeling to explore interactions with sleep health, burnout, and mental health [46], as for sleep duration [7]. Such studies could lead to the identification of organizational characteristics that are detrimental or beneficial to sleep and workplace well-being, enabling recommendations for organizational changes. At the same time, individual factors such as mental health and socioeconomic status are also crucial contributors to burnout. The implementation of support systems, including psychological evaluation, could facilitate early diagnosis and thus improve the prognosis of sleep alterations and their consequences [8].

Previous studies have involved the exploration of relationships between separate sleep health dimensions and mental health. The RU-SATED scale dimensions of “Regularity” and “Satisfaction” had the strongest estimated associations with depression, and the “Timing” and “Efficiency” dimensions had the weakest associations [47]. The inclusion of night schedule data in our models may explain the lack of association between “Regularity” and burnout. A post hoc analysis showed that “Regularity” was associated with burnout among night workers (*n* = 199). The “Satisfaction”, “Alertness”, and “Efficiency” dimensions of the Spanish version of the SATED scale (the RU-SATED instrument without “Regularity”) correlated strongly with anxiety and depression [17]. In another assessment of the same five sleep health dimensions, “Satisfaction” and “Alertness” had the strongest estimated longitudinal associations with incident depression and “Duration” had the weakest association [22]. Overall, the “Satisfaction” and “Alertness” dimensions appear to be important for mental health, as suggested in a study conducted to validate the French version of the RU-SATED scale [18]. Thus, they could be used to assess sleep health in relation to burnout. The discrepancy in association strength among the six sleep health dimensions suggests the need for a better understanding of the concept of sleep health and the distinction between sleep quality dimensions (“Alertness” and “Satisfaction”) and sleep behavior dimensions (“Regularity”, “Timing”, and “Duration”) [18]. Each dimension (except for “Satisfaction”) can also be measured objectively [48] and each is related to physiological processes that may have different implications for health [14]. Indeed, “Regularity”, “Timing”, “Efficiency”, and “Duration” showed the strongest associations with mortality, whereas “Satisfaction” was borderline significant and “Alertness” was not significant in a study of 2887 men followed during 11 years [49]. This point is important because behavioral sleep interventions have different efficacies, depending on the dimension targeted [50]. Thus, the relationships of sleep health dimensions to other burnout dimensions need to be investigated. Indeed, as described by the job demand–control model, personal accomplishment could compensate for situations with high levels of EE and depersonalization [51]. Future studies should assess the relationship of sleep health to the other two dimensions of burnout (depersonalization and personal accomplishment) using a suitable questionnaire.

### 4.3. Limitations

This study has several limitations. First, participants were included on a voluntary basis, leading to a low response rate (13.0%) and a probable over-representation of persons interested in sleep. However, the sample was representative of Bordeaux University Hospital HCWs in terms of age (mean, 42.5 years in the Bordeaux University Hospital population and 39.2 years in our sample), sex (83.3% and 84.2% female), and job category (87.7% and 81.0% nurses). Moreover, the mean RU-SATED score (7.9) was comparable to that from an international study conducted during the COVID-19 lockdown (8.1) [32]. The rates of anxiety and depressive symptoms were high (35.9% and 15.6%, respectively), but comparable to those reported for the post–COVID-19 lockdown period (27.8% and 26.9%, respectively) [52]. Except for mental health, there were no effect-modifying factors for the association between sleep health and burnout in the literature and no interactions were found in our data. Therefore, a selection bias that could alter the association between sleep health and burnout was unlikely. The lack of representativeness of this sample should not hinder the interpretation of associations between variables of interest [53]. Second, objective sleep measurements were not taken in this study. Thus, the presence or absence of sleep disorders (e.g., obstructive sleep apnea syndrome, chronic insomnia disorder, and restless legs syndrome) were not assessed, although they constitute risk factors for poor sleep heath and burnout. Further studies should explore the relationship between sleep health and burnout while considering sleep disorders in order to distinguish general and clinical population. Third, we used a single-item scale to assess EE, a proxy for burnout, which may seem to be insufficient to identify burnout with precision. However, the scale has been validated [33,34], and the burnout rate among nurses in our sample (13.4%) is consistent with that obtained in an international meta-analysis (11.2%) [4]. Moreover, mental health assessment did not rely on a clinical diagnosis of anxiety and depression disorders, but used a short self-reported scale (PHQ-4) that is a valid and reliable tool to evaluate the severity of the anxiety and depressive symptoms. Fourth, the study was conducted during the COVID-19 health crisis with a population of HCWs with increased psychological distress. Indeed, sleep health and workplace well-being were disturbed during this period [11,13]. However, working in a COVID-19 unit had no impact on the results of this study, probably because it was conducted at the end of the first lockdown in a region relatively spared by the outbreak and in a period with few health restrictions and no saturation of the healthcare system. Replication of this study should confirm that the findings can be extrapolated to other health contexts. Fifth, the study was limited by its cross-sectional design. No causal inference can be made between burnout and sleep health. Due to the entanglement of many associated factors in this relationship, longitudinal approaches using structural equation modeling based on documented mechanistic assumptions are needed.

## 5. Conclusions

Although the cross-sectional design of this study hampered the drawing of causal inferences, our findings suggest: (1) that sleep health, and in particular the “Satisfaction” and “Alertness” RU-SATED dimension scores, are associated with HCW burnout; (2) that sleep health alteration could constitute a risk factor of HCW burnout; and (3) that sleep health promotion should be investigated as a strategy to reduce HCW burnout and could be used to complement work organization–centered strategies.

## Figures and Tables

**Figure 1 jcm-12-01895-f001:**
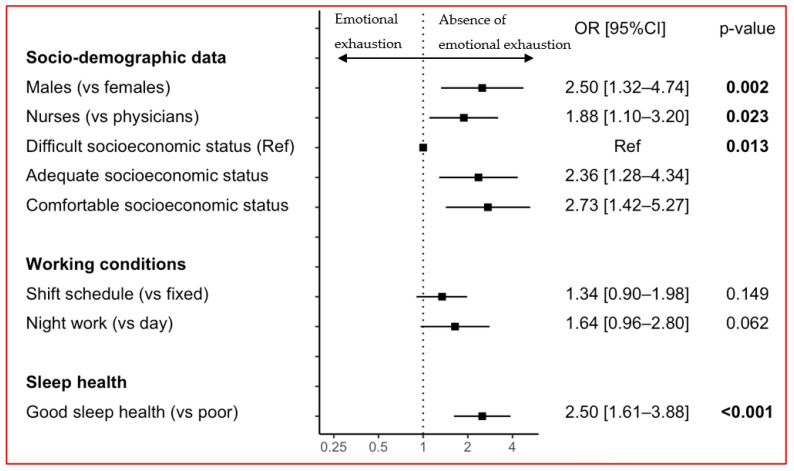
Adjusted associations between the absence of emotional exhaustion and socio-demographic data, working condition, and sleep health. Forest plot of the associations with the absence of emotional exhaustion after adjustment for age, sex, job category, perceived socioeconomic status, work schedule, night work, telecommuting, work in a COVID-19 unit, self-perceived health, and sleep health.

**Figure 2 jcm-12-01895-f002:**
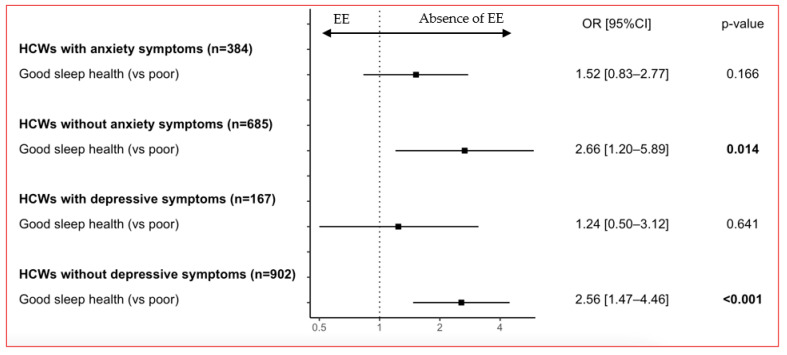
Adjusted associations between sleep health and the absence of emotional exhaustion (EE) according to anxiety and depressive symptoms. Forest plot of the associations with the absence of emotional exhaustion (EE) after adjustment for age, sex, job category, perceived socioeconomic status, work schedule, night work, telecommuting, work in a COVID-19 unit, self-perceived health, and sleep health, stratified for anxiety and depressive symptoms.

**Figure 3 jcm-12-01895-f003:**
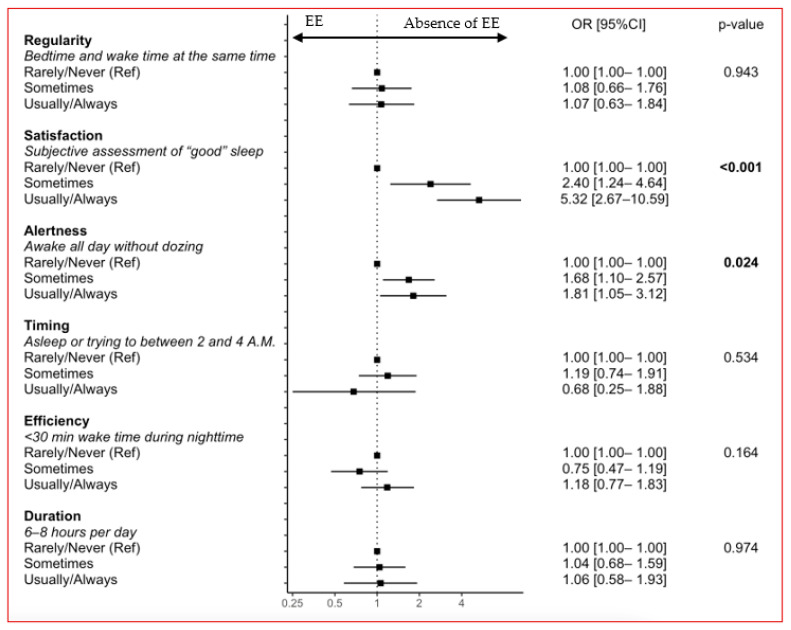
Adjusted associations between each sleep health dimension and the absence of emotional exhaustion (EE). Forest plot of the associations with the absence of emotional exhaustion (EE) after adjustment for age, sex, job category, perceived socioeconomic status, work schedule, night work, telecommuting, work in a COVID-19 unit, self-perceived health, and sleep health.

**Table 1 jcm-12-01895-t001:** Characteristics of the study population and their distributions according to sleep health and emotional exhaustion.

	Total *n* = 1069		Sleep Health ^a^			Emotional Exhaustion ^b^			
		Good *n* = 474	Poor *n* = 595	Statistic	*p*-Value	Yes *n* = 143	No *n* = 926	Statistic	*p*-Value
Age (years)	39.2 ± 11.3	39.8 ± 10.8	39.8 ± 11.7	t = 1.41	0.159	40.4 ± 10.8	39.1 ± 11.4	t = 1.39	0.167
Sex - Male - Female	169 (15.8%) 900 (84.2%)	80 (16.9%) 394 (83.1%)	89 (15.0%) 506 (85.0%)	X^2^ = 0.593	0.393	12 (8.4%) 131 (91.6%)	157 (17.0%) 769 (83.0%)	X^2^ = 6.20	0.009
Job category - Nurses - Physicians	866 (81.0%) 203 (19.0%)	332 (70.0%) 142 (30.0%)	534 (89.8%) 61 (10.2%)	X^2^ = 65.3	<0.001	116 (81.1%) 27 (18.9%)	750 (81.0%) 176 (19.0%)	X^2^ < 0.001	0.972
Perceived socioeconomic status - Comfortable - Adequate - Difficult	501 (46.9%) 501 (46.9%) 67 (6.3%)	287 (60.6%) 172 (36.3%) 15 (3.2%)	214 (36.0%) 329 (55.3%) 52 (8.7%)	X^2^ = 67.4	<0.001	56 (39.2%) 67 (46.9%) 20 (14.0%)	445 (48.1%) 434 (46.9%) 47 (5.1%)	X^2^ = 17.8	<0.001
Work schedule: - Shift - Fixed	514 (48.1%) 555 (51.9%)	192 (40.5%) 282 (59.5%)	322 (54.1%) 273 (45.9%)	X^2^ = 19.0	<0.001	67 (46.9%) 76 (53.2%)	447 (48.3%) 479 (51.7%)	X^2^ = 0.051	0.821
Night work: - Yes - No	199 (18.6%) 870 (81.4%)	50 (10.6%) 424 (89.4%)	149 (25.0%) 446 (75.0%)	X^2^ = 35.6	<0.001	22 (15.4%) 121 (84.6%)	177 (19.1%) 749 (80.9%)	X^2^ = 0.905	0.342
Telecommuting: Yes	168 (15.7%)	81 (17.1%)	87 (14.6%)	X^2^ = 1.03	0.271	24 (16.8%)	144 (15.6%)	X^2^ = 0.064	0.706
Work in a COVID-19 unit: Yes	234 (21.9%)	90 (19.0%)	144 (24.2%)	X^2^ = 3.90	0.041	30 (21.0%)	204 (22.0%)	X^2^ = 0.030	0.777
Self-perceived health: Poor	163 (15.3%)	32 (6.8%)	131 (22.0%)	X^2^ = 46.4	<0.001	44 (30.8%)	119 (12.9%)	X^2^ = 29.4	<0.001
Anxiety symptoms: Yes	384 (35.9%)	98 (20.7%)	286 (48.1%)	X^2^ = 84.8	<0.001	110 (76.9%)	274 (29.6%)	X^2^ = 119	<0.001
Depressive symptoms: Yes	167 (15.6%)	34 (7.2%)	133 (22.4%)	X^2^ = 45.0	<0.001	65 (45.5%)	102 (11.0%)	X^2^ = 109	<0.001
Sleep health: Good	474 (44.3%)					38 (26.6%)	436 (47.1%)	X^2^ = 20.3	<0.001
Emotional exhaustion: Yes	143 (13.4%)	38 (8.0%)	105 (17.7%)	X^2^ = 20.3	<0.001				<0.001

^a^ Measured by the RU-SATED with good sleep health defined as a score > 8; ^b^ measured by the single-item version of the Maslach Burnout Inventory with emotional exhaustion defined as a frequency of once a week or more; the student *t*-test was used for continuous variables, while the chi-square test was used for categorical variables to test the univariate associations of each characteristic with good sleep health and the absence of emotional exhaustion.

## Data Availability

Not applicable.

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
