# Peer review of "Emotional Exhaustion, a Proxy for Burnout, Is Associated with Sleep Health in French Healthcare Workers without Anxiety or Depressive Symptoms: A Cross-Sectional Study"

_jcm, 2023, doi:10.3390/jcm12051895_

Round 1

Reviewer 1 Report

Dear authors,

Congratulations for your very interesting article, you have done a real great work. I have only few minor remarks/questions:

1. did you obtain some ethical approval for your research?

2. at line 30 it is not clear the formulation, "the end of the first 2019 coronavirus disease–associated lockdown in France" - the first lockdown has started in March 2020, right?

3. I'm not sure if it is readible clear enough the sentence from 244-245 "Thus, improvement of the sleep health of individuals without sleep disorders may benefit their workplace well-being" - why only for individuals without sleep disorders? For the others (with sleep disorder) the improvement of the sleep health doesn't have a positive impact on their workplace well-being? Maybe a little bit extensions of the results/discussion/conclusions will make your paper more clear.

Everything else is very good, congrats!

Reviewer 2 Report

This is a very interesting topic adding to the current discourse on whether burnout is a form of depression or a phenomenon on its own.

Line 108-116. Depression and anxiety disorders cannot and should not be diagnosed by self-report scales such as PHQ-4 which is meant to be used for screening only. I suggest changing the title of the article to reflect that these results are about HCWs "below the screening threshold" or "with low scores on PHQ-4", or it can be worded as "after controlling for self-reported depressive and anxiety symptoms". Otherwise it can be seen as misleading.

Line 129. Why was a single item used as a proxy for burnout? Was this by design or simply because participants only provided answers to it? This should be made clearer. Similarly to above, it is better practice not to use a proxy but to state in the title and discussion what exactly was correlated with what, e.g. "emotional exhaustion was correlated with sleep health", rather than "burnout" which includes more than the single item.

Line 134. It would be better not to split the Maslach results on emotional exhaustion into "yes/no" burnout categories. It is best to correlate emotional exhaustion scores with scores on the sleep questionnaire and use PHQ-9/PHQ-4 scores as covariates. Similarly, use raw sleep scores to correlate, instead of splitting into categories.

Best to use a regression model, rather than a series of chi-square tests.

Line 165. This is a very low response rate. What were the reasons for it? Was there sampling bias? This should be clearer in the discussion, as well as what the effects of this bias would be on the results of the study and why.

Table 1. You report large and highly significant demographic differences. Were these controlled for? These results should be explicitly discussed in the discussion section. 

Figure 2. State which variables were the results adjusted for.

Were multiple comparison adjustments used for the t-tests and chi-square tests?

Round 2

Reviewer 2 Report

Thank you for addressing my concerns. I wish you further successes in this difficult field.